# Clinical, Histological, Cytogenetic and Molecular Analysis of Monozygous Twins with Wilms Tumor

**DOI:** 10.3390/genes13020372

**Published:** 2022-02-18

**Authors:** Katarzyna Iwańczyk, Bartosz Czachowski, Patrycja Sosnowska-Sienkiewicz, Gabriela Telman, Paulina Ciążyńska, Przemysław Mańkowski, Danuta Januszkiewicz-Lewandowska

**Affiliations:** 1Medical Centrum NASMEDICA, 63-000 Środa Wielkopolska, Poland; iwanczyk.katarzyna@onet.pl; 2Specialist Health Care Center for Mother and Child, 61-825 Poznan, Poland; bartosz.czachowski@gmail.com; 3Department of Pediatric Surgery, Traumatology and Urology, Poznan University of Medical Sciences, 60-572 Poznan, Poland; mankowskip@ump.edu.pl; 4Department of Pediatric Oncology, Hematology and Transplantology, Poznan University of Medical Sciences, 60-572 Poznan, Poland; gabriela.telman@gmail.com (G.T.); danuta.januszkiewicz@ump.edu.pl (D.J.-L.); 5Department of Medical Diagnostics Poznan, 60-595 Dobra, Poland; p.ciazynska@zdmdobra.pl

**Keywords:** child, genetics, kidney neoplasms, monozygous twins, oncology, Wilms’ tumor

## Abstract

The familial occurrence of childhood cancers has been proven for a long time. Wilms’ tumors often do not have a clear germline genetic cause. However, approximately 2% of all nephroblastoma cases are familial. Descriptions of twins with the same cancer are extremely rare, so our aim was to present the background of the available literature of the occurrence of Wilms’ tumor in a pair of monozygotic twin girls with detailed clinical, histological, and molecular analysis. Two twins were born of unrelated Caucasian parents. Family history revealed no known chronic diseases or malformations. At the age of 3.5 years, the first twin was admitted to the emergency department due to hematuria and abdominal pain. Ultrasound examination revealed an enlarged right kidney, 12.8 cm, with a mass in the upper pole measuring 56 × 69 × 78 mm. The second girl was referred for an abdominal ultrasound, which revealed a right kidney measuring 8.6 cm with a central mass measuring 54 × 45 × 41 mm. Both children underwent surgical resection, and the histopathological result showed a mixed form of nephroblastoma, predominantly epithelioid with residual blastemal compartment. Detailed clinical, histological, cytogenetic, and molecular analyses were performed on both sisters. It was also decided to identify environmental factors. Information was obtained that the girls’ parents run a farm and regularly use pesticides and chemical rodenticides. Based on our observations and the available literature, Wilms tumor in monozygotic twins may be present. Both genetic and environmental factors may be involved in the development of tumors. After excluding methylation abnormalities and mutations in the genes studied, we questioned whether the onset of Wilms tumor in both sisters could be the result of exposure of the twins’ parents to pesticides.

## 1. Introduction

The familial occurrence of childhood cancers has been proven for a long time [1]. One of the most well-known inherited cancer is leukemia. For instance, familial acute lymphoblastic leukemia (ALL) was confirmed in siblings or dizygotic twins [2], and retinoblastoma was described among family members with confirmed RB1 gene mutation [3].

Afshar and Golden described a case of monozygotic twins with neuroblastoma [4]. Wilms’ tumor (WT) often does not have a clear germline genetic cause. However, approximately 2% of all nephroblastoma cases are familial [5]. The detection of chromosomal abnormalities on chromosome 11p in cases of WAGR syndrome eventually led to the localization and identification in 1991 of the WT1 gene, a gene involved in the pathogenesis of Wilms tumor [6].

Although it has been described that the risk of WT is reduced in multiples [6], a few cases of concurrent disease in twins are known. The first case of identical twins with Wilms tumor was described by Gaulin in 1951 [7]. Svane presented another pair of children with nephroblastoma, in which diagnosis of WT in one child occurred 3 months after the detection of WT in the first one [8]. Perotti et al. focused on a comprehensive molecular analysis of monozygotic twins with WT [9]. Cases of non-twin siblings diagnosed with WT at different ages are also known in the literature. These were the offspring of unrelated [10], as well as related parents [11].

Descriptions of twins with the same cancer are extremely rare, so our aim was to present the background of the available literature of the occurrence of WT in a pair of monozygotic twin girls with detailed clinical, histological, and molecular analysis.

## 2. Case Report

Two twins were born of unrelated Caucasian parents. Family history revealed no known chronic diseases or malformations. The twins were from the mother’s first pregnancy and were delivered by cesarean section at 36 weeks gestation. A single placenta was found.

The first twin had Apgar scores of 8-10-10 and a birth weight of 1940 g. At birth, the neonate appeared normal, without any disease or malformation. On day 5 of life, supraventricular tachycardia was diagnosed, which required pharmacological treatment. At 12 months of age, the child required no further cardiac care or treatment. Her psychomotor development was within normal limits. At the age of 3.5 years, the patient was admitted to the emergency room due to hematuria and abdominal pain. Ultrasound examination revealed an enlarged right kidney with a mass in the upper pole measuring 10 × 9 × 12 cm (Figure 1). Chest X-ray showed metastases to both lungs. A computed tomography (CT) scan excluded central nervous system lesions confirmed a tumor in the right kidney and multiple metastatic nodules in both lungs. Based on the clinical picture and imaging studies, Wilms’ tumor was diagnosed in stage IV, and the neoadjuvant chemotherapy according to the SIOP protocol was given [12,13]. After 6 weeks of chemotherapy, a significant reduction in the tumor mass in the kidney was observed in CT scan. Lung metastases were still present, although smaller. A right-sided radical nephroureterectomy was performed, and treatment was continued according to the protocol. In the 11th week of postoperative treatment, the control CT examination of the lungs did not reveal any metastases, so the lung radiotherapy was abandoned. The histopathological result showed a mixed form of nephroblastoma, predominantly epithelioid with residual blastemal compartment, intermediate risk, local stage II (Figure 2). The child is currently 12 months from completion of treatment and remains in full remission of the disease.

The second twin at birth had an Apgar score of 8-9-9 (no crying, then irregular) with a body weight of 1760 g. The child’s development was normal. Due to the Wilms tumor found in her sister, 3 days after the sister’s diagnosis, the girl was referred for an abdominal ultrasound, which revealed a right kidney with a central mass measuring 7 × 6 × 8 cm (Figure 1). CT scan showed metastatic subpleural nodules in the right lung. At the time of ultrasound and CT, the girl was asymptomatic, and the tumor was not palpable on physical examination. A diagnosis of Wilms tumor stage IV was made, and similar treatment to her sister was given. The second twin girl underwent an uncomplicated right radical nephroureterectomy and did not require radiotherapy to the lungs (follow-up CT scan did not show metastases). She completed treatment 1.5 weeks later after her sister. The histopathological examination was identical to that of her sister (Figure 2). The girl, like her sister, is now in full remission of the disease.

Both sisters received preoperative chemotherapy according to SIOP protocol for stage IV. It lasted 6 weeks and included administration of six doses of VCR 1.5 mg/m^2^ every week, three doses of actinomycin 45 ug/kg every 2 weeks, and two doses of doxorubicin 50 mg/m^2^ at week 1 and 5. Tumor’s size on CT examination decreased before and after 6 weeks of chemotherapy, respectively, in the first twin from 10 × 9.3 × 12.6 cm to 5.2 × 4.5 × 6.7 cm, and in the second twin from 7 × 6.1 × 7.7 cm to 3 × 4 × 2.8 cm (Figure 1). In both sisters, mixed form of Wilms tumor, intermediate-risk group, local stage II were diagnosed. Hence, radiotherapy to the abdominal cavity was not used. Postoperative chemotherapy included a 4-drug regimen according to SIOP protocol and lasted 34 weeks.

Both sisters underwent karyotype (was normal) and NGS testing as well as examination for imprinting loci. NGS using Agilent’s SureSelect XT Target Panel, which includes a set of probes for the analysis of 220 genes with documented clinical relevance to oncologic disorders, was performed on an Illumina NexSeq 550 instrument. Analysis conducted using Variant Studio v.3.0, and IGV v.2.3 software did not show the presence of pathogenic single nucleotide changes. Analysis of imprinted DNA methylation was performed at the following loci: DIRAS3 (1p31); PLAGL1 (6q24); GRB10 (7p12); PEG1/MEST (7q32); KCNQ1OT1/H19/IGF2 DMRO (11p15); DLK1 (14q32); SNRPN (15q11), PEG3 (19q32); and NESPAS/GNAS (20q13). DNA from both sisters showed no deviation from normal methylation levels at any of these loci.

## 3. Discussion

Familial occurrence of Wilms tumor is very rare and usually affects siblings and first cousins. This fact was first described by Gaulin in 1951. A pair of monozygotic twins were operated on at 15 months of age [7]. Subsequently, Hewitt et al. and Puumala et al. published similar cases [1,6]. Wilms tumor in a pair of monozygotic twins was described in few papers and occurred once in both or only in one of them [8,9,10,11,14,15]. Olson et al. described the occurrence of Wilms tumor in one twin and medulloblastoma in the other in two pairs of monozygotic twins. In the first pair, congenital anomalies were present in both patients, but a tumor was found in only one child. In the second pair, the tumor was present in both children, but congenital anomalies were found in only one of them [15]. In the twins we treated, we did not observe any congenital anomalies but only the simultaneous occurrence of Wilms tumor.

The occurrence of the tumor in both twin sisters, the early age of onset, and the identical histology of the tumors may suggest the involvement of genetic factors, such as the WT1 gene, in the etiology of the disease in our girls. Several studies have suggested an association between bilateral WT, early diagnosis, and germline mutations in the WT1 gene [5,16,17,18]. Most commonly, alterations in the WT1 gene have been observed in children with Wilms tumor and associated congenital anomalies, although cases are also known where Wilms tumor, as in our girls, is not accompanied by any congenital anomalies [18,19]. No mutations in the WT1 gene were found in our patients.

Although the accurate reason for the origins of the same tumors at a similar time in multiples is not well known yet, there are two hypotheses explaining that phenomenon. The first theory suggests that tumors in twins arise from transplacental spread [20,21]. In a publication by Shatar et al., such a mechanism was suggested in a case of twins with neuroblastoma, where one twin developed a primary tumor, and the other had metastatic disease without a clear primary location. In this study, the molecular profile of embryonal neuroblastoma in monozygotic twins is presented. For each twin, comparative genomic hybridization was performed in liver and peripheral blood samples. The signature of copy number changes in the tumors of both twins was similar and suggested a common clonal origin. Additional findings included large deletion of chromosome 10 and amplification of chromosome 17 [20].

The second assumption indicates that the twins share the same genetic abnormality that causes the tumor, nephroblastoma [9]. In the case of the twins described by Perotti et al., analyses performed on blood samples and on tumor sections did not reveal karyotype abnormalities [9]. No mutations in the WT1 gene were identified in the constitutional and tumor DNA of the twins. In addition, in the normal kidney of one of them, no abnormalities in WT1 expression were found. However, loss of heterozygosity on chromosome 11p, involving alleles of maternal origin, was found in both the single tumor of twin no. 1 and in two separate tumors of twin no. 2, suggesting a common etiology of these diseases. Chromosomal abnormalities and single-gene mutations have been confirmed in other reported cases of hereditary WT as well [18,19].

Detailed analysis of NGS and methylation abnormalities performed in our patients did not clearly confirm either of the two theories mentioned above. Therefore, an environmental factor began to be suspected. Information was obtained that the girls’ parents run a farm and regularly use pesticides and chemical rodenticides. Based on their study, Rios et al. suggested an association between Wilms tumor incidence and maternal exposure to household pesticides during pregnancy [22]. These conclusions were also confirmed by Cooney et al. in their publication [23]. In the latter analysis, it was shown that pre-conceptional exposure to pesticides may play a role in the development of Wilms tumor by causing damage to germinal stem cells. Chu et al. also described in their publication the effect of pesticide exposure in parents on cancer development in their children. An association between pesticide exposure and the development of Wilms tumor, childhood leukemia, brain cancer, neuroblastoma, and Ewing’s sarcoma has been suggested [24]. All authors emphasize the need for future prospective or cohort studies to help better understand the association between pesticide exposure and the risk of developing Wilms tumor [22,23,24]. After excluding methylation abnormalities and mutations in the genes studied, we questioned whether the onset of Wilms tumor in both sisters could be the result of exposure of the twins’ parents to pesticides. Although previous studies have reported that the risk of Wilms’ tumor in the second twin is reduced despite the presence of Wilms’ tumor in the first twin [1,5], the girls we presented indicate the importance of performing anticipatory testing even in asymptomatic siblings.

## 4. Conclusions

In conclusion, based on our observations and the available literature, Wilms tumor in monozygotic twins may be present. Both genetic and environmental factors may be involved in the development of tumors, including WT. Wilms tumor may be caused by increased exposure to pesticides and rodenticides in the parents of the affected child.

## Figures and Tables

**Figure 1 genes-13-00372-f001:**
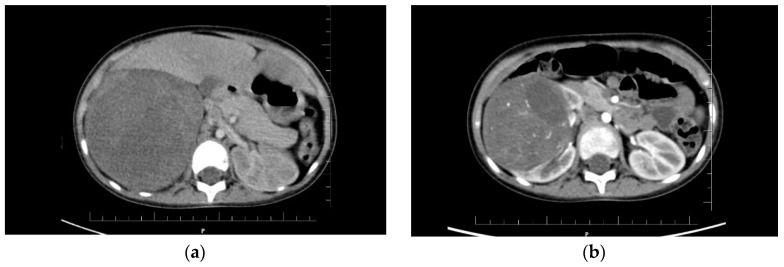
The results of the computed tomography examination in both patients: (**a**) Visible tumor of the right kidney in the first girl, size 10 × 9.3 × 12.6 cm; (**b**) Visible tumor of the right kidney in the second child, size 7 × 6.1 × 7.7 cm.

**Figure 2 genes-13-00372-f002:**
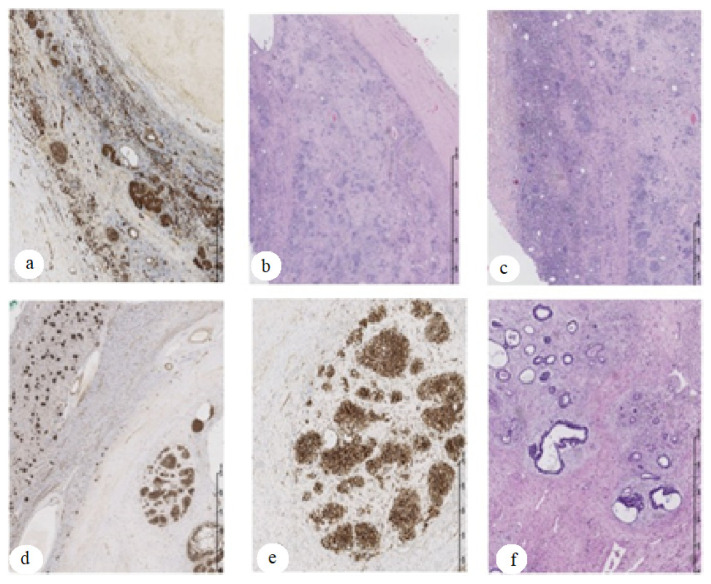
Wilms tumor mixed type with intermediate risk. Tumor specimen examination after 6-week preoperative chemotherapy. HE staining showed residual blastemic-epithelial compartment with regression areas. Immunohistochemistry with WT1 antibody showed residual blastemic nodules with epithelial elements; slides (**a**–**c**)—the tumor tissue sections of the first sister; slides (**d**–**f**)—the tumor tissue sections of the second sister. Slides (**b**,**c**,**f**)—HE staining, magnification ×20. Slides (**a**,**d**,**e**)—WT1 antibody (Origene) immunohistochemistry analysis in formalin-fixed and paraffin-embedded tumor tissue followed by peroxidase conjugation of the secondary antibody and DAB staining; magnification ×20 for slides (**a**) and (**d**) and ×40 for slide (**e**).

## Data Availability

Data available on request due to restrictions.

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
