# Peer review of "Clinical, Histological, Cytogenetic and Molecular Analysis of Monozygous Twins with Wilms Tumor"

_genes, 2022, doi:10.3390/genes13020372_

Round 1

Reviewer 1 Report

This case report details the clinical, histological, cytogenetic, and molecular analysis of monozygous twins with Wilms tumor. The case is very interesting; however, I feel that important details and analyses are missing which are really needed to maximize our learning from the current case presentation.

  1. Somatic tumor tissue analysis really needs to be performed to answer the question about the clonal origin of the tumors, especially for publication in a genetics journal. Did the tumor metastasize from one twin to another? This could be answered with whole exome analysis or SNP array comparing tumor tissue.
  2. The photograph of the two children is not informative; no abnormal phenotype such as hemihypertrophy, abnormal facies etc is being described and so I think the photograph is unnecessary. 
  3. A figure with the histologic appearance and immunohistochemical staining used to characterize the Wilms tumor specimens from each child is needed. WT1 staining, others?
  4. Critical clinical details are missing from the report. What neoadjuvant chemotherapy was given and for how long before surgical resection? Was there a volumetric response in the tumor during neoadjuvant chemotherapy? What was the local stage and the SIOP risk histology for each resected tumor (low, intermediate, high)? What adjuvant chemotherapy was given and for how long in each case? What adjuvant radiation therapy if any was given in each case? Some of these details are stated for one twin but not the other. 
  5. The word "grade" is used instead of stage in the  manuscript in multiple places. This will be very confusing for the reader and should be corrected. 
  6. The point about rodenticide and pesticides being the potential cause for the monozygous twins getting wilms tumor is important, but somewhat speculative. I feel like this statement and conclusion should be more carefully stated in the discussion section.
  7. I don't agree that the "pathogenesis of Wilms tumor usually lacks clear genetic factors." I think the somatic genetic factors are well-described. There is actually also an expanding body of knowledge about the germline predisposing factors also. Maybe the authors should rephrase this to say Wilms tumor often does not have a clear germline genetic cause.
  8. The abandonment of lung radiotherapy does not seem to adhere to the SIOP protocol. Can the authors explain this better or note that this was an individualized decision of the treating team rather than a protocol-driven choice?
  9. What assay was used to detect alterations in DNA methylation at imprinted loci? What was the source of DNA for the germline studies?
  10. Minor points:
    1. Abstract line 30 - would change to say "Both children underwent surgical resection and the histopathological result..."
    2. Line 73: would change to say "The first twin had Apgar scores of 8-10-10 and a birth weight of 1940g."
    3. Line 80-81 - should say CT scan instead of CK
    4. Line 86 - Would refer to the operation as a radical nephroureterectomy instead of nephrectomy.
    5. Line 99 - Says "grade IV" when I think the authors mean "stage IV". This occurs in several places in the manuscript. 

Author Response

Dear Reviewers,

Dear Editor,

Thank you very much for your valuable remarks and constructive comments. We have addressed them point by point. Additionally, in the revised version of the manuscript, changed or added sections of text have been marked in yellow.

We are submitting a revised manuscript entitled "Clinical, Histological, Cytogenetic and Molecular Analysis of Monozygous Twins with Wilms Tumor. Case Report" and we hope that the revised version will be found satisfactory for publication in Genes.

Kind regards,

Danuta Januszkiewicz-Lewandowska

Patrycja Sosnowska-Sienkiewicz

Reviewer 1.

This case report details the clinical, histological, cytogenetic, and molecular analysis of monozygous twins with Wilms tumor. The case is very interesting; however, I feel that important details and analyses are missing which are really needed to maximize our learning from the current case presentation.

  1. Somatic tumor tissue analysis really needs to be performed to answer the question about the clonal origin of the tumors, especially for publication in a genetics journal. Did the tumor metastasize from one twin to another? This could be answered with whole exome analysis or SNP array comparing tumor tissue.

Re: Thank you for this idea. The girls developed Wilms tumor at the age of 3 years, and after all, Wilms tumor grows very rapidly, within a several weeks, but not years. Hence, we do not think that the origin of the tumor can be linked to metastasis from one sister to another during intrauterine life. Certainly, whole genome or SNP array analysis of tumor tissue could yield new information. However, we feel that these analyses would not confirm the clonal origin of the tumor cells.

2. The photograph of the two children is not informative; no abnormal phenotype such as hemihypertrophy, abnormal facies etc is being described and so I think the photograph is unnecessary.

Re: Thank you for this comment. We agree that the girls don't show any abnormal phenotype, so the photo has been removed.

3. A figure with the histologic appearance and immunohistochemical staining used to characterize the Wilms tumor specimens from each child is needed. WT1 staining, others?

Re: Thank you for this comment - we agree that presenting histopathologic images of both sisters' tumors would be educational. These are attached to the manuscript text as Fig.2.

3. Critical clinical details are missing from the report. What neoadjuvant chemotherapy was given and for how long before surgical resection? Was there a volumetric response in the tumor during neoadjuvant chemotherapy? What was the local stage and the SIOP risk histology for each resected tumor (low, intermediate, high)? What adjuvant chemotherapy was given and for how long in each case? What adjuvant radiation therapy if any was given in each case? Some of these details are stated for one twin but not the other.

Re: Both sisters received preoperative chemotherapy according to SIOP protocol for stage IV - it lasted 6 weeks and included administration of 6 doses of VCR 1.5 mg/m^2 every week, 3 doses of Actinomycin 45ug/kg every 2 weeks and two doses of Doxorubicin 50mg/m^2 at week 1st and 5th. Tumor’s size on CT examination decreased before and after 6 weeks of chemotherapy, respectively in first twin from 10x9,3x12,6 cm to 5,2x4,5x6,7cm, and in second twin from 7x6,1x7,7 cm to 3x4x2,8cm.  In both sisters mixed form of Wilms tumor, intermediate risk group, local stage II was diagnosed. Hence, radiotherapy to the abdominal cavity was not used. Postoperative chemotherapy included 4-drug regimen according to SIOP protocol and lasted 34 weeks.

All these sentences were added to the manuscript.

4. The word "grade" is used instead of stage in the  manuscript in multiple places. This will be very confusing for the reader and should be corrected. 

Re: Thank you for that rightful remark. Yes, we of course agree - this is our mistake. We have inserted the correct word "stage" everywhere in the text of the manuscript.

5. The point about rodenticide and pesticides being the potential cause for the monozygous twins getting wilms tumor is important, but somewhat speculative. I feel like this statement and conclusion should be more carefully stated in the discussion section.

Re: Thank you for this comment. We agree that this paragraph should be written with softer message. Hence, we have changed this sentence either in abstract, as well as in discussion in the version as below.

After excluding methylation abnormalities and mutations in the genes studied, we questioned whether the onset of Wilms tumor in both sisters could be the result of exposure of the twins' parents to pesticides.

6. I don't agree that the "pathogenesis of Wilms tumor usually lacks clear genetic factors." I think the somatic genetic factors are well-described. There is actually also an expanding body of knowledge about the germline predisposing factors also. Maybe the authors should rephrase this to say Wilms tumor often does not have a clear germline genetic cause.

Re: Thank you for this valuable comment. We agree that there is a growing amount of knowledge about genome-wide predisposing factors for Wilms tumor. The proposed rephrased version of our sentence is definitely an improvement. Thank you again. This sentence has been revised both in the abstract and in the text of the manuscript.

7. The abandonment of lung radiotherapy does not seem to adhere to the SIOP protocol. Can the authors explain this better or note that this was an individualized decision of the treating team rather than a protocol-driven choice?

Re: According to the SIOP protocol for intermediate risk histology, there is no indication for lung radiotherapy when there is complete remission of the lung metastases after chemotherapy (week 10 of postoperative chemotherapy) as assessed by CT scan or surgery. And that was exactly the situation for both sisters.

8. What assay was used to detect alterations in DNA methylation at imprinted loci? What was the source of DNA for the germline studies?

Re: Peripheral blood lymphocytes were the source of DNA for germline studies. Similarly, peripheral blood was the starting material for detection of DNA methylation changes at the imprinted loci. Regards to the study of methylation disorders, we collaborate with Wessex Regional Genetics Laboratory, where we perform imprinting studies.

  1. Minor points:

1. Abstract line 30 - would change to say "Both children underwent surgical resection and the histopathological result..."

Re: Thank you for this comment. It was corrected.

2. Line 73: would change to say "The first twin had Apgar scores of 8-10-10 and a birth weight of 1940g."

Re: Thank you for this comment. It was corrected.

3. Line 80-81 - should say CT scan instead of CK

Re: Thank you. It was corrected.

4. Line 86 - Would refer to the operation as a radical nephroureterectomy instead of nephrectomy.

Re: Thank you for this comment. This was changed to "radical nephroureterectomy" in the descriptions of both patients.

5. Line 99 - Says "grade IV" when I think the authors mean "stage IV". This occurs in several places in the manuscript.

Re: Thank you for that rightful remark. Yes, we of course agree - this is our mistake. We have inserted the correct word "stage" everywhere in the text of the manuscript.

Reviewer 2 Report

Very interesting case report

Some comments:

1. the images of Computer tomography (CT) and pathology pictures are important, could authors provide these information? 

Author Response

Dear Reviewers,

Dear Editor,

Thank you very much for your valuable remarks and constructive comments. We have addressed them point by point. Additionally, in the revised version of the manuscript, changed or added sections of text have been marked in yellow.

We are submitting a revised manuscript entitled "Clinical, Histological, Cytogenetic and Molecular Analysis of Monozygous Twins with Wilms Tumor. Case Report" and we hope that the revised version will be found satisfactory for publication in Genes.

Kind regards,

Danuta Januszkiewicz-Lewandowska

Patrycja Sosnowska-Sienkiewicz

Reviewer 2.

Very interesting case report

Some comments:

  1. the images of Computer tomography (CT) and pathology pictures are important, could authors provide these informations? 

Re: Thank you very much for this valuable comment. We agree that adding imaging documentation will make the manuscript more attractive. We have attached the abdominal CT findings of both girls to the publication (Figure 1).

Thank you for this comment - we agree that presenting histopathologic images of both sisters' tumors would be educational. These are attached to the manuscript text as Figure 2.

Figure 2. Wilms tumor mixed type with intermediate risk. Tumor specimen examination after 6-week preoperative chemotherapy. HE staining showed residual blastemic-epithelial compartment with regression areas. Immunohistochemistry with WT1 antibody showed residual blastemic nodules with epithelial elements; slides (1-3)- the tumor tissue sections of the first sister; slides (4-6)- the tumor tissue sections of the second sister. Slides (2-3 and 6)- HE staining, magnification x20. Slides (1, 4-5)- WT1 antibody (Origene) immunohistochemistry analysis in formalin fixed and paraffin embedded tumor tissue followed by peroxidase conjugation of the secondary antibody and DAB staining; magnification x20 for slides 1 & 4 and x40 for slide 5.

Round 2

Reviewer 1 Report

The reviews are satisfactory and I recommend that the manuscript be published.